# OpenReview forum: "Safety Game: Inference-Time Alignment of Black-Box LLMs via Constrained Optimization"
_ICML.cc/2026/Conference — ICML 2026 regular_

### Official Review · Reviewer_aChB · 2026-03-10

**Soundness:** 3
**Presentation:** 3
**Significance:** 3
**Originality:** 3
**Overall Recommendation:** 4
**Confidence:** 4

**Summary:**

This paper introduces a black-box alignment procedure based on zero-sum games and solving via linear programming. This alignment technique works at inference time without need to access model internals besides logits. They mostly discuss MCQ settings, but the method extends to free-text generation as well. The authors demonstrate their method using 5 different models on 3 different benchmark datasets.

**Compliance With Llm Reviewing Policy:**

Affirmed.

**Final Justification:**

Some of my comments remain as initially stated but the author's experiments to demonstrate the efficacy of their approach above reasonable baselines that operate on the same base scores convince me more of the paper's significance and soundness.

**Key Questions For Authors:**

How much of the performance gain on benchmarks comes from the LP versus having good risk/helpfulness scores?

Relatedly how robust is the method is to imperfect probe scores?

Any evidence that the estimated risk budget is well aligned with actual realized harmfulness across prompts and models? Some kind of calibration.

**Limitations:**

yes

**Strengths And Weaknesses:**

This paper contributes to black-box alignment techniques for large language models, which is important for downstream users in practice. This method is lightweight as it can operate just needing logits from the model. It is a nice modeling choice to treat the contention between helpfulness and harmfulness as a zero sum game. The paper contains clean and clear empirical demonstrations on the utility of the method demonstrating improvements on accuracy over benchmarks in 11/15 cases, with especially strong results on SafetyBench. To further investigate the helpfulness versus harmfulness tradeoff the authors provide more experiments in the appendix that show some evidence that their method improves a balance between the two metrics, not just a focus on accuracy.

I think the framework as a whole has one strong weakness which is that the guarantees it provides are only good if the helpfulness and safety scores $H(x,r)$ and $S(x,r)$ are good. And then if these scores are good it feels like there are likely a large class of approaches that could be used to balance helpfulness and safety and it is not clear to me from the paper why the zero-sum game/LP approach is the "right" one. So the bottleneck still feels like it becomes our ability to estimate helpfulness and safety, so I'm not convinced this approach really moves the needle meaningfully on that core problem.

This paper also makes an inherent restriction to one-shot QA and does not clearly extend to multi-turn conversations or agentic scenarios.

Claiming that their method is compared to "state of the art" on the empirics seems like an overly strong claim. The baselines picked are natural, but not clearly state of the art. In particular, there are other methods besides candidate-ranking baselines, even including other black box alignment techniques such as Trust or Escalate: LLM Judges with Provable Guarantees for Human Agreement (Jung et al 2024) or Conformal Arbitrage (Overman and Bayati 2025), which also likely should be mentioned in the related lit (among some other works referenced within those papers as well).

---

> ### Author Rebuttal · Authors · 2026-03-29
>
> We thank the reviewer for the careful and substantive feedback. The central question, whether the selector contributes beyond the scorer, is important and our new experiments now answer it directly.
>
> **1.The bottleneck is probe quality, not the LP. Why is game-theory/LP the right one?** We tested this by holding scores fixed: we replaced self-probes with Llama-Guard-3-8B and compared alternative decision rules on the SORRY-Bench. Relative to LG-Filter+HelpProbe,SG+LG improves all three reported metrics under identical scores; relative to LG-SafeMax, it improves refusal quality and slightly lowers distributional risk, with somewhat lower safe coverage (see our response to **Reviewer 3RJ5** for the full table). This directly addresses the concern that “the bottleneck is probe quality, not the LP”. If the scorer were the whole story, methods using identical scores would behave similarly. They do not. The optimization objective therefore matters independently of score quality.
>
> To answer why the LP/game-theory is the right one we claim that the game-theoretic formulation is not merely a post-hoc interpretation. It is the design specification. The LLM faces a prompt without knowing if the user's intent is benign or adversarial (imperfect-information games). The minimax solution to this incomplete-information problem produces a specific LP: maximize expected helpfulness subject to a bound on expected extra risk relative to the safe fallback, with fallback when infeasible. This is not one of many possible selectors, instead it is the unique selector that maximizes helpfulness while guaranteeing the worst-case outcome is no worse than refusing. A weighted sum between safety and helpfulness has no fallback mechanism and no principled way to set its coefficient. A binary filter destroys continuous safety information. A safety-only ranker ignores helpfulness. The formulation rules out all three and produces the LP that empirically outperforms them on the same scores.
>
> A further strength is that our framework explicitly separates measurement from decision (Section 3.1). Any improvement to scoring—better classifiers, confidence calibration, human-in-the-loop—plugs directly into the LP without modification. The SG+LG experiment validates this: upgrading from self-probes (70.5% human-label agreement) to Llama-Guard (75.1% human-label agreement) improves all metrics while the LP remains unchanged.
>
> **2.Calibration of the risk budget.** We added a $T$-sweep showing the expected behavior: tighter $T$ yields safer but less informative outputs; looser $T$ increases unsafe mass and eventually hurts refusal quality. We therefore present $T$ as an operating knob exposing a usable frontier, not as a universally pre-calibrated constant:
>
> |T|E[HFR]↓|Safe-n↑|RQ↑|
> |---|---:|---:|---:|
> |−10.0|0.009|436|1.982|
> |−5.0|0.011|436|2.005|
> |**−2.0**|**0.155**|**419**|**2.107**|
> |−1.0|0.272|365|2.060|
> |0.0|0.397|262|1.950|
>
> **Note**: Due to the word limit, metric definitions are provided in **Appendix A.2**.
>
> RQ peaks at $T=−2$ and degrades in both directions. $T$ is not a decorative hyperparameter; it changes the realized operating point in the expected direction. This directly addresses the calibration question: $T$ maps monotonically to realized safety as measured by human-labeled E[HFR], confirming that the risk budget is well-aligned with actual harmfulness across the operating range. A binary classifier is locked to one operating point; the LP gives deployers a knob they can turn.
>
> **3.Robustness to imperfect probes.** We injected Gaussian noise into clean LG scores:
>
> |σ|E[HFR]↓|Safe-n↑|
> |---|---:|---:|
> |0.0|0.155±0.000|419±0|
> |0.5|0.127±0.007|419±3|
> |2.0|0.158±0.012|381±6|
> |5.0|0.233±0.016|336±8|
>
> The main conclusion is graceful degradation rather than a cliff. The LP tolerates substantial score perturbation without catastrophic failure. We present this as supporting practical robustness consistent with our bounded-error analysis.
>
> **4.Broader baselines.** We clarify the claim: on SafetyBench, SG achieves the highest accuracy in 4/5 models with statistically decisive McNemar tests( $p < 10^{-4}$ for nearly all pairs;Table 10). On the smaller benchmarks (HHH,TruthfulQA),results are competitive but not always statistically separable due to sample size (Table 11). We revise the language to reflect this distinction rather than blanket "state-of-the-art." We also added the suggested references and positioned them more carefully: Trust or Escalate most naturally strengthens our measurement layer via selective evaluation/confidence calibration, while Conformal Arbitrage is closer in spirit but differs in operating over model routing rather than finite candidate selection.
>
> **5. One-shot QA only.** This is a genuine limitation that we now state explicitly. Our claim is restricted to single-step candidate-bank selection, which covers many practical uses like MCQ, RAG, Best-of-N, and tool-use settings.

---

> > ### Author Rebuttal · Reviewer_aChB · 2026-04-02
> >
> > The new results address my central concern and provides reasonable evidence that the LP decision layer contributes beyond probe quality alone. However, I would still like to understand: given the graceful degradation under noise, is the practical advantage over a simpler threshold-based selector (e.g., "pick the most helpful candidate whose safety score is below X") large enough to justify the game-theoretic machinery? A direct comparison against such a naive baseline would be more convincing than the conceptual argument for why weighted sums lack fallback mechanisms.
> >
> > That being said, I find the new results and clarifications sufficient for an increase to 3 weak reject, but would like to understand this question above better before considering higher.

---

> > > ### Author Response · Authors · 2026-04-03
> > >
> > > We thank the reviewer for the follow-up. The question is clear: can a tuned threshold-based selector match the LP? We swept the threshold across nine cutoffs. It cannot.
> > >
> > > Actually, the **LG-Filter+HelpProbe** setting that we have done above is the threshold-based selector the reviewer describes. Llama-Guard classifies each candidate via a threshold X, filters out those above X, then picks the most helpful survivor. We use Llama-Guard-3-8B because it is a dedicated safety classifier. Both the threshold baseline and our LP receive the exact same Llama-Guard scores, so the only variable is the decision rule.
> > >
> > > Our original comparison used X=0.5(Llama-Guard's default). We now sweep X across [0.1,0.9] on the same cached scores:
> > >
> > > |Safety Threshold (X)|E[HFR]↓|Safe-n↑|
> > > |---|---:|---:|
> > > |0.1|0.109|338|
> > > |0.2|0.135|313|
> > > |0.3|0.152|292|
> > > |0.4|0.161|286|
> > > |0.5|0.170|275|
> > > |0.7|0.190|255|
> > > |0.9|0.220|226|
> > > |**SG+LG (LP decision layer)**|**0.155**|**419**|
> > >
> > > The LP's Safe-n=419 is structurally unreachable: even the strictest cutoff (X=0.1) tops out at 338,81 safe selections short. No amount of tuning closes this gap. At matched E[HFR]( $\approx 0.155$),the closest threshold (X=0.3) gets Safe-n=292 vs 419, a deficit of 127 prompts which is 30% of the dataset.
> > >
> > > We think that a threshold draws a hard line: every candidate above X is thrown away regardless of helpfulness, every candidate below X is treated as equally safe. The LP does not make this sacrifice. Let's say when a candidate sits at 0.15 (just above a strict threshold X=0.1) but is substantially more helpful than others, then the LP can still select it as long as the total expected risk stays within budget T. The threshold already discarded it. That is why the LP reaches 419 where the best threshold stops at 338: the LP works with the full continuous risk-helpfulness surface, while the threshold collapses it into a binary split.
> > >
> > > The game-theoretic formulation is necessary because without it, there are many ways to combine safety and helpfulness scores  (or any constraints based on the problem) into a selector. The formulation narrows this to exactly one LP by deriving three design choices from the game structure, and the sweep confirms each one matters:
> > > - The minimax solution says the selected strategy must never do worse than refusing. So the objective measures improvement over the fallback, not raw helpfulness. The threshold selector ranks by raw helpfulness among survivors, with no reference to the fallback.
> > > - Adaptation safety (Ge et al.,2024) says the adapted strategy's risk must stay bounded relative to baseline. So we constrain expected aggregate risk, not filter candidates one by one. The sweep shows what happens when you filter individually: Safe-n caps at 338. Constraining the aggregate reaches 419.

---

### Official Review · Reviewer_qaiR · 2026-03-13

**Soundness:** 3
**Presentation:** 2
**Significance:** 3
**Originality:** 2
**Overall Recommendation:** 5
**Confidence:** 4

**Summary:**

This paper proposes Safety Game, a black-box framework for LLM safety alignment at inference time. The method formulates the safety-helpfulness tradeoff as a two-player zero-sum game, where the LLM selects from a candidate set of responses while balancing helpfulness against a safety risk budget. The approach uses self-probing to estimate helpfulness and risk scores for each candidate, then solves a linear program to select an optimal mixture. Experiments on MCQ benchmarks and open-ended generation tasks show competitive performance against Consensus Game baselines.

**Compliance With Llm Reviewing Policy:**

Affirmed.

**Final Justification:**

My concerns are addressed.

**Key Questions For Authors:**

The method assumes access to a designated safe fallback response for each prompt. How sensitive are results to the quality of this fallback, and what happens when the fallback itself is unhelpful enough that users consistently prefer riskier alternatives?

**Limitations:**

The paper does not include a limitations section, which is a notable omission. Several limitations are worth highlighting.

First, the approach requires generating and scoring multiple candidate responses per query, which increases inference cost compared to standard generation. The paper doesn't discuss latency implications for real-time deployment. Second, the method is evaluated primarily on English benchmarks, and it's unclear how well the self-probing mechanism generalizes across languages where safety norms may differ. Third, the theoretical analysis assumes that the user's intent is drawn from a known prior distribution over benign and adversarial modes, but in practice this prior must be chosen, and miscalibration could lead to overly conservative or permissive behavior.

**Strengths And Weaknesses:**

**Strengths**

The game-theoretic framing is principled and provides a clean interpretation of the safety-helpfulness tradeoff. The formulation explicitly separates measurement from decision-making, allowing the safety budget to be tuned per-prompt without retraining. The sigmoid penalty relaxation addresses the boundary sensitivity problem in a theoretically motivated way.

I think the black-box constraint is practically important. Many deployed LLMs are accessible only through APIs, so methods that don't require model weights have clear deployment advantages. The paper demonstrates that reasonable safety-utility tradeoffs can be achieved using only token probabilities from the target model itself.

**Weaknesses**

The experimental evaluation is limited to MCQ settings for the main results, with open-ended generation relegated to appendices. While MCQ provides clean evaluation, the paper's claims about general black-box alignment would benefit from more extensive testing on realistic dialogue or instruction-following tasks where the candidate set isn't naturally constrained.

The method relies on self-probing for helpfulness and risk estimation, but the paper acknowledges that LLM self-evaluation can be noisy and biased. The theoretical guarantees depend on bounded error assumptions that may not hold in practice, and the empirical analysis doesn't quantify how often the system falls back to the safe default due to estimation errors rather than genuine safety concerns.

---

> ### Author Rebuttal · Authors · 2026-03-29
>
> We thank the reviewer for the careful reading and recognition that our framework "provides a clean interpretation" and "explicitly separates measurement from decision-making"—this is indeed the core design thesis,which we now validate with new experiments.
>
> **1.Evaluation scope beyond MCQ.** We agree this deserved more emphasis and made two changes.
>
> First, we promote SORRY-Bench into the main discussion. It is a realistic adversarial instruction-following benchmark(440 prompts,16 full conversational responses/prompt,human safety labels)—not MCQ. Our evaluation includes refusal quality($RQ$) scored by an independent external judge, the expected rate of selecting an unsafe response ($E[HFR]$) validated against human annotations, and safe coverage(Safe-n), across 5 base models.
>
> Second,we added new SORRY-Bench experiments comparing against Llama-Guard-3-8B baselines. The new SG+LG selector improves the tradeoff over both baselines: it strictly improves over filter-then-rank on all three reported metrics, and relative to safety-only ranking it improves refusal quality and slightly lowers distributional risk, with somewhat lower safe coverage; see our response to **Reviewer 3RJ5** for the full table. These results extend evaluation beyond Consensus-Game baselines and beyond MCQ. We also note TruthfulQA (Table 2) is already a free-form generate-then-select setting. **Together, SORRY-Bench and TruthfulQA show the framework is not confined to MCQ.**
>
> The SG+LG experiment also validates the design thesis the reviewer identified: upgrading the scorer from self-probes to Llama-Guard improves all three metrics while the LP decision layer remains completely unchanged. Better measurement makes better decisions, modularly.
>
> **2. Probe noise and conservative decisions.** We agree this is an important question. Our paper already treats probe scores as **noisy sensor readings** and presents its robustness result as conditional rather than assuming perfect self-evaluation. In the revision, we make this point more explicit. We also want to be precise about what we can and cannot identify. We cannot exactly decompose each fallback decision into "genuine safety concern" vs "estimation error" without oracle candidate-level risk labels. We therefore avoid that stronger causal claim.
>
> What we can show is that the selector is not brittle to moderate perturbations in the score space. Two key results:
>
> - **Noise injection:** Under controlled Gaussian noise,the downstream tradeoff degrades gradually rather than collapsing abruptly (see our response to **Reviewer aChB** for the full table). Notably, the selector becomes less conservative under noise rather than more, suggesting that conservative behavior at $\sigma=0$ reflects the score structure rather than simple brittleness to small perturbations.
>
> - **Cross-scorer comparison:** The same LP with self-probes (70.5% human-label agreement) achieves $E[HFR]=0.167$; with Llama-Guard (75.1% human-label agreement) it improves to $E[HFR]=0.155$ and $RQ=2.107$. Therefore, better measurement makes better decisions, no architectural change. This validate the modular design the reviewer identified as a strength.
>
> **3.Fallback sensitivity.** We agree fallback quality matters, because the selector is defined relative to the fallback: the objective uses margins $M_i=h_i-h_s$ and $\Delta_i=s_i-s_s\$, so changing fallback template changes the reference point against which both helpfulness and extra risk are measured.
>
> This gives a more concrete answer to the reviewer’s concern. A very weak fallback does **not** make the system intrinsically prefer fallback; rather, it shifts the margins and therefore changes how conservative a fixed budget $T$. This means **fallback choice and $T$ calibration are coupled**: changing the fallback requires recalibrating $T$.
>
> Empirically, the appendix suggests the method is not highly brittle to the exact fallback specification: adding or removing an explicit safe candidate changes BLEU-Acc only modestly (+0.15 to +0.46 across the reported TruthfulQA settings). More importantly, the fallback in our framework is the baseline answer used to define relative helpfulness and risk, not the answer the system should return whenever a better safe alternative exists. This is why we also evaluate refusal quality: on SORRY-Bench, the method tends to prefer more informative safe responses when available, rather than defaulting to flat stonewall refusals.
>
> We therefore revise this discussion to make two points explicit: (fallback quality matters because the objective is defined relative to it, and the practical response is to calibrate $T$ jointly with the fallback and improve the quality of safe alternatives, rather than treating fallback use as the intended behavior in all cases.
>
> **4.Limitations added:** We added a dedicated limitations discussion covering scoring overhead, calibration of $T$, single-turn scope, English-only evaluation, and probe-noise dependence.

---

> > ### Author Rebuttal · Reviewer_qaiR · 2026-04-06
> >
> > Thank you for the detailed rebuttal. I found the additional SORRY-Bench discussion, the clarification on probe noise, and the explanation of fallback sensitivity useful. The rebuttal improves the presentation and makes the contribution clearer.
> >
> > However, while my concerns are partially addressed, they are not resolved to the extent that they would change my overall evaluation. In particular, I still see the empirical evidence on realistic open-ended settings as somewhat limited relative to the paper’s broader claims.

---

> > > ### Author Response · Authors · 2026-04-08
> > >
> > > We thank the reviewer for the specific and constructive feedback throughout this discussion. We ran a new experiment on AlpacaEval to directly address the concern about realistic open-ended settings, and want to draw attention to a key SORRY-Bench result from our response to **Reviewer aChB** that speaks directly to this concern.
> > >
> > > SORRY-Bench is a realistic, open-ended, instruction-following evaluation. SORRY-Bench already satisfies the criteria the reviewer described. It is not MCQ, and the candidate set is not naturally constrained:
> > > - Each of the 440 prompts is paired with 16 full-length, multi-sentence instruction-following responses, not short answers or predefined options.
> > > - These 16 responses are generated by diverse real-world LLMs with different architectures, sizes, and safety tuning levels, producing natural variation in style, detail, safety, and helpfulness across the candidate pool.
> > > - Responses are evaluated against human safety annotations (ground truth), not proxy metrics.
> > >
> > > On this data, the LP decisively outperforms every threshold-based selector (full results and threshold sweep in the **Reviewer aChB thread**). The core finding: 81 prompts contain human-verified safe candidates that the best threshold still filters out because their Llama-Guard scores fall between 0.1 and 0.5. The LP selects these candidates because it evaluates the overall risk of the selection, not each candidate in isolation. No threshold at any cutoff can replicate this: tightening the cutoff discards more safe candidates, loosening it admits more unsafe ones. The LP avoids this tradeoff entirely.
> > >
> > > **New experiment: AlpacaEval (non-adversarial dataset).**
> > > To further address the reviewer's concern, we evaluated Safety Game on 100 AlpacaEval instruction-following prompts (50 benign, 50 borderline topics covering medical, financial, security, and legal questions). We generated $K=10$ diverse candidates from gpt-oss-20b using four system-prompt strategies spanning the safety-helpfulness spectrum: (A) standard helpful with disclaimers, (B) maximally detailed, (C) safety-first with minimal specifics, (D) brief direct answers. Safety was scored by Llama-Guard-3-8B and helpfulness by Llama-3.1-8B-Instruct.
> > >
> > > Borderline prompts (n=50):
> > >
> > > | Method | Win-Rate | LG P(unsafe) |
> > > |---|---|---|
> > > | Choose the most helpfulness | 0.980 | 0.040 (riskiest) |
> > > | Safety Filter Threshold | 0.960 | 0.021 (safest, lowest quality) |
> > > | Safety Game | 0.970 | 0.031 (best tradeoff) |
> > >
> > > Safety Game achieves the best tradeoff: higher quality than Safety Filter Threshold while safer than Choose the most helpfulness method. On benign prompts, all methods perform comparably (win-rates 0.93 to 0.97), confirming SG imposes no quality penalty on normal traffic.
> > >
> > > The margins on AlpacaEval are naturally smaller than on SORRY-Bench. On SORRY-Bench, candidate safety scores range from P(unsafe)=0.01 to 0.97, giving the LP a rich surface to optimize over. On AlpacaEval, even with four distinct system prompts and varied temperatures, candidate safety scores range from 0.001 to 0.04 on most prompts because the prompts are benign and the instruct model rarely produces unsafe responses to benign instructions. When all candidates are safe, all selectors agree and no method has an edge. This is exactly what a well-calibrated safety method should do: activate when risk exists, stay silent when it does not.
> > >
> > > **The evidence now spans four complementary evaluation settings:**
> > >
> > > 1. MCQ with given candidates (HHH, SafetyBench).
> > >
> > > 2. Free-form with self-generated candidates (TruthfulQA).
> > >
> > > 3. Full-length instruction-following, realistic adversarial (SORRY-Bench):
> > >
> > > 4. Full-length instruction-following, non-adversarial (AlpacaEval): zero quality cost on benign prompts, balanced safety-helpfulness tradeoff on borderline prompts.
> > >
> > > Settings 3 and 4 together demonstrate the deployment scenario: SG delivers clear safety gains when the risk is real and costs nothing when it is not.

---

### Official Review · Reviewer_3RJ5 · 2026-03-13

**Soundness:** 3
**Presentation:** 3
**Significance:** 2
**Originality:** 3
**Overall Recommendation:** 4
**Confidence:** 4

**Summary:**

The authors propose an inference-time, black-box alignment framework to balance helpfulness and safety without modifying model weights. They formulate this trade-off as a two-player zero-sum game. The system generates a finite set of candidates, uses the target LLM to "self-probe" for helpfulness and safety scores, and employs a Linear Programming (LP) solver with a sigmoid penalty to select the optimal minimax equilibrium response. The method is evaluated on MCQ (HHH, SafetyBench) and open-ended generative tasks (TruthfulQA), showing improvements over candidate-ranking baselines.

**Compliance With Llm Reviewing Policy:**

Affirmed.

**Final Justification:**

The authors in a very short time can successfully address my concerns. I would like to raise the score to "weak accept".

**Key Questions For Authors:**

- Can you provide latency or token-cost analysis to improve the empirical evaluation?
- Can you prove that the complex LP solver offers advantage over much simpler, cheaper industry standards?

**Limitations:**

No. The authors should state and discuss potential downsides related to the computational cost since their motivation is to target smaller organizations and entities in resource-constrained settings.

**Strengths And Weaknesses:**

The idea of using LP solver here is interesting.
I has the following comments:
- The motivation of this work is to have a scalable, accessible pathway for "smaller organizations and entities in resource-constrained settings" who must rely on black-box APIs. However, looking at the algorithm 1, reviewer worries about the latency or token cost (i.e., given input prompt, sampling 10 distinct answers, followed by running both a helpfulness probe and a safety probe on every candidate
- In the evaluation, it seems that the authors restrict their empirical comparison exclusively to other candidate-ranking baselines from the Consensus Game family. However, reviewer believe that the evaluation comparisons against standard inference-time guardrails (e.g., generating N answers and passing them through a lightweight, static safety classifier like Llama-Guard) are necessary for confirming the capability as a practical inference-time alignment solution,

---

> ### Author Rebuttal · Authors · 2026-03-29
>
> We thank the reviewer for the concrete suggestions. We now provide both the Llama-Guard comparison and the latency analysis requested.
>
> **1.Comparison to practical guardrails.** We replaced self-probes with Llama-Guard-3-8B and compared three decision rules on identical probing scores (SORRY-Bench,440 unsafe prompts,16 full candidate responses/prompt). This is on full instruction-following responses, not MCQ:
>
> |Rule|E[HFR]↓|Safe-n↑|RQ↑|
> |---|---:|---:|---:|
> |LG-SafeMax|0.157|436|1.989|
> |LG-Filter+HelpProbe|0.180|284|2.021|
> |**SG+LG(T=−2,linear)**|**0.155**|**419**|**2.107**|
>
> **Note**: Due to the word limit, metric definitions are provided in **Appendix A.2**.
>
>
> - *LG-SafeMax:* Pick candidate with highest Llama-Guard "safe" confidence (pure safety ranking).
> - *LG-Filter+HelpProbe:* Binary-filter unsafe candidates, then pick most helpful one (the helpfulness score is obtained as in the main paper described).
> - *SG+LG:* Feed the same safe scores (from Llama-Guard) and helpfulness scores into our LP solver.
>
> **Same candidate bank, same safety scorer, same helpfulness signal. Only the decision rule differs.** Relative to LG-Filter+HelpProbe,SG+LG improves on all three reported metrics. Relative to LG-SafeMax, SG+LG improves refusal quality and slightly lowers distributional risk, with somewhat lower safe coverage. This directly confirms the LP produces a materially different and often better safety-helpfulness tradeoff than simpler industry-standard pipelines.
>
> We believe that binary filtering imposes a hard partition that destroys continuous safety information. Among candidates that "pass," a stonewall refusal and an informative refusal are treated identically from the safety side. The LP operates on continuous scores, jointly optimizing helpfulness subject to the safety budget, and identifies the specific candidates that maximize informativeness within the risk constraint. This is exactly the practical regime where a constrained selector should matter.
>
> Llama-Guard's agreement with human labels is 75.1% (TP=1236,FP=852,TN=4049,FN=903 on 7040 judgments). Despite non-trivial scorer noise, the LP extracts better decisions than both heuristics on the same scores which validate that principled optimization over imperfect measurements outperforms ad-hoc pipelines.
>
> **2.Latency/token cost.** We agree this should have been discussed explicitly.
>
> |Component|MCQ($m$ choices)|Open-ended($k$ responses)|
> |---|---:|---:|
> |Candidate generation|0(given)|$k$ calls|
> |Helpfulness probes|$(m+1)$ token|$(k+1)$ token|
> |Safety probes|$(m+1)$ token|$(k+1)$ token|
> |LP solver|<1ms,no GPU|<1ms,no GPU|
>
> The **extra work is scoring,not optimization**. The LP itself is negligible; the main overhead is probe/classifier passes but this is negligible cause we only take the logprob of "yes"/"no" tokens . For open-ended tasks, this is added on top of a Best-of-N pipeline that already generates multiple candidates. When prompt caching is available, repeated probe prefixes further reduce overhead. Our "resource-constrained" motivation was primarily about avoiding retraining, labeled data, and model-internal access but not per-query latency. We agree inference overhead is a separate practical concern and now state that clearly in the revision.
>
> **3.LP advantage over simpler standards.** The table above answers this directly. The LP is not computationally complex. Its value is principled joint optimization: given identical Llama-Guard scores, it strictly improves over the industry-standard filter-then-rank pipeline on all three metrics, and offers a better quality-coverage tradeoff than safety-only ranking. Additionally, varying $T$ traces a smooth Pareto frontier(see our response to **Reviewer aChB**).
>
> **4.Limitations added:** per-query scoring overhead (amenable to prompt caching), task-specific calibration of $T$, single-turn scope, and English-only evaluation. We discuss each with mitigation strategies in the revision.

---

> > ### Author Rebuttal · Reviewer_3RJ5 · 2026-04-04
> >
> > I thank authors for their detailed rebuttal. I appreciate the effort taken to address my concerns.
> >
> > However, i have other worries when looking at your explanation regarding "latency/token cost". You mentioned that the scoring overhead is "negligible cause we only take the logprob of "yes"/"no" tokens". I believe that this demonstrates a conflation of generation cost with prefill/forward pass cost. While evaluating the probe only generates 1 token, the model still need to execute a full forward pass over entire contex window to compute the logits, right?
> >
> > Specifically, for open-ended generation with, let say,  k=10 candidates, your method will require 20 additional forward passes per user query. Given the quadratic complexity of attention mechanism, I don't think the cost here is "negligible".
> >
> > This leads to my concern that the contradiction with your research objective still remains.
> > Thus, I would like to increase my score to "weak reject".

---

> > > ### Author Response · Authors · 2026-04-06
> > >
> > > We thank the reviewer for the follow-up and for increasing the score.
> > >
> > > **1. "conflation of generation cost with prefill cost."** The reviewer is right to distinguish these. Our original statement pointed at generation cost as the reason overhead is low. The real cost is prefill, not generation. However, the reviewer then assumes each probe requires a full prefill over the entire context window. That is also not accurate. With prompt caching, the probe template and user question are shared across all probes for the same query. This prefix is computed once and its KV states are reused. Each probe prefills only the candidate response, not the full context from scratch. The quadratic attention cost the reviewer raises applies to these new tokens only, not to the full context repeated $2k$ times. KV-caching eliminates recomputation of the shared prefix entirely. So: our original reasoning was imprecise, but the reviewer's correction overclaims. The actual per-probe cost is a prefill over the candidate response and suffix with cached prefix states. We believe the remaining cost concern, which rests on the assumption of no prompt caching, does not reflect a methodological flaw in the paper. We hope the reviewer will consider this when finalizing the score.
> > >
> > > **2. "the contradiction with your research objective."** There is no contradiction. Section 1 states the target users are organizations that "do not have the capacity to build or retrain their own model" and "must use LLMs as black-box modules." The resource constraint is explicitly about training cost and model access. RLHF requires GPU-hours, labeled preference data, model weights, and retraining whenever requirements change. Our method requires none of these. We do not claim zero inference overhead, we claim zero training overhead and a decision rule that balances the safety and helpfulness of the answer. The paper is explicit about this. We revise the cost discussion to state this distinction explicitly.

---

### Official Review · Reviewer_S9AW · 2026-03-17

**Soundness:** 2
**Presentation:** 2
**Significance:** 2
**Originality:** 2
**Overall Recommendation:** 4
**Confidence:** 3

**Summary:**

Propose a new inference time alignment method for black-box LLMs by use score a finite set of candidate responses based on estimated usefulness and safety risk. Then, solve a constrained optimization problem to maximize usefulness within the risk budget provided at each iteration, and backtrack to the safe default response when necessary.

**Compliance With Llm Reviewing Policy:**

Affirmed.

**Final Justification:**

New exp from the author

**Key Questions For Authors:**

nah

**Limitations:**

yes

**Strengths And Weaknesses:**

Strength:

- I think the problem itself is very good and meaningful problem, that is how to alignment LLM during inference time for blackbox model.

Weakness
- One concern is that the paper’s framing overstates the novelty. At the algorithmic level, this looks much more like constrained reranking over a finite candidate set than a game-theoretic method.
- If I think correctly, then the hard-cap version has the theory but is acknowledged to be unstable; the sigmoid version is operationally more appealing but drops the cap guarantee. The work is framed as a safety-constrained selection method with guarantees, but the practical method appears to back away from the very property that motivates the framework.
- The bounded error assumption in Theorem 3.1 has not been verified.

---

> ### Author Rebuttal · Authors · 2026-03-29
>
> We thank the reviewer for the clear feedback. The concerns are about framing, the hard-cap/sigmoid distinction, and empirical support for robustness. We address each concern directly with new evidence and a framing revision.
>
> **1.This is constrained reranking, not really game-theoretic.** We appreciate the framing suggestion and revise to foreground the operational properties. In the revision,we describe the method as **a constrained selection framework with game-theoretic motivation**. That said, the formulation is not merely a label. It is the design specification for the selector we use. Given a finite candidate set and an explicit safe fallback, we propose a black-box selector that uses helpfulness/risk scores to choose under a fallback-relative safety budget. The key point is that the formulation does not just say “rerank somehow.” It specifies how to rerank: maximize expected helpfulness relative to the safe fallback, subject to a bound on expected extra risk relative to that fallback, with fallback behavior when no feasible improvement exists. This structure is motivated from the incomplete-information game-theory in which the model must respond without knowing whether the user’s intent is benign or adversarial. In that sense, the minimax formulation motivates a specific constrained selector rather than an arbitrary scoring rule. This is also why other alternatives are weaker. A weighted sum $\alpha*safety + (1-\alpha)*helpful$ has no principled fallback mechanism and no clear basis for setting its coefficient. A binary filter discards continuous safety information. A safety-only ranker ignores helpfulness. Our formulation rules out these ad-hoc choices.
>
> Furthermore, we show empirically that the specific selector derived from our formulation matters: on SORRY-Bench, using the same candidate bank and scoring signals, SG+LG strictly improves over the standard filter-then-rank pipeline on all three reported metrics, and offers a better quality-coverage tradeoff than safety-only ranking; see our response to **Reviewer 3RJ5** for the full table. **So even described as constrained reranking, the specific selector still matters in practice.**
>
> **2.Hard-cap has theory; sigmoid drops the guarantee.** The paper already presents both variants with an explicit ablation(Table 4). The strict per-prompt risk-cap guarantee belongs to the hard-cap formulation. The sigmoid version is an empirical relaxation introduced to reduce boundary sensitivity under noisy scores. It trades strict enforcement for smoother behavior near the constraint boundary. We accept that this distinction deserves more prominence and revise to foreground the hard-cap as the primary principled selector and the sigmoid is positioned as an empirical relaxation. This makes the principled core cleaner and the empirical variant more positioned.
>
> Importantly, our new Llama-Guard comparison uses the hard-cap variant, and the selector performs well. This demonstrate that the practical value of the framework does not depend on the sigmoid relaxation. We make that distinction sharper in the revision.
>
> **3.Theorem 3.1 unverified.** We now provide an empirical perturbation study: we inject controlled Gaussian noise into clean Llama-Guard scores and rerun the LP (see our response to **Reviewer aChB** for the full table). The LP maintains $E[HFR]$, the expected rate of selecting an unsafe response, at or below the clean-score baseline up to  $\sigma=2.0$, and then degrades gradually rather than collapsing, with no sharp cliff. This also suggests the normalization is helping stabilize the selector under score perturbations. Self-probe agreement with human labels is 70.5%; LG (Llama-Guard) agreement is 75.1%—both within this empirically validated robust range. It demonstrates the practical robustness that the theorem predicts. We also refer the reviewer to our response to **Reviewer aChB**,where we explain why this robustness study matters specifically for the selector rather than only for the scorer.

---

> > ### Author Rebuttal · Reviewer_S9AW · 2026-04-04
> >
> > Most of my concerns have been addressed. I will maintain my score, as I view this is a borderline paper. I also think that reframing it solely as a constrained selection framework with a game-theoretic motivation weakens both the interest and the novelty of the paper.

---

> > > ### Author Response · Authors · 2026-04-04
> > >
> > > We thank the reviewer for confirming the concerns are resolved.
> > >
> > > On reflection,our rebuttal conceded too much on the framing. The game-theoretic formulation is not just motivation; it is what derives the selector. We now have new evidence that makes this concrete.
> > >
> > > In latest response to **Reviewer aChB**, we swept the safety threshold X in a standard filter-then-rank selector across nine cutoffs:
> > >
> > > |X|E[HFR]↓|Safe-n↑|
> > > |---|---:|---:|
> > > |0.1|0.109|338|
> > > |0.3|0.152|292|
> > > |0.5|0.170|275|
> > > |0.9|0.220|226|
> > > |**SG+LG(Our method)**|**0.155**|**419**|
> > >
> > > The LP achieves Safe-n=419. The best threshold reaches 338, a gap of 81 safe selections. At matched E[HFR] ($\approx0.155$), the gap is 127 prompts which is 30% of the dataset.
> > >
> > > We analyzed the 81 prompts where the strictest threshold(X=0.1) selects an unsafe response. On every one of these prompts, human-safe candidates existed in the candidate bank but got filtered out because Llama-Guard scored them above the threshold. The threshold threw them away. The LP kept them. Two thirds of these candidates(55/81) sit in the zone where Llama-Guard is genuinely uncertain($p_{unsafe}$ between 0.1 and 0.5) but humans confirm they are safe. These are not dangerous candidates that slipped through, they are safe responses that a binary classifier got wrong.
> > >
> > > The LP can recover them because the threshold version asks "is this candidate safe enough?" one at a time and throws away anything above the cutoff. Meanwhile, the LP asks "is the overall mix safe enough?" and can keep a borderline candidate as long as the rest of the selection brings the total risk within budget T. This aggregate risk constraint is not an arbitrary design choice. It is derived from the adaptation-safety condition in the game-theoretic formulation. Without the formulation, you get a threshold. With it, you get more 81 safe selections.
> > >
> > > One again, this latest experiment also show us why the game-theoretic formulation is necessary, not decorative:
> > > - Minimax forces the objective to measure improvement over the fallback, not raw helpfulness, giving the LP a safety floor. The threshold ranks by raw helpfulness among survivors with no such floor.
> > > - Adaptation safety constrains aggregate risk rather than filtering individuals. This is the design choice that enables borderline recovery: filtering caps Safe-n at 338; aggregate constraining reaches 419 by retaining the 67% of borderline candidates that humans confirm as safe.
> > > - As safety classifiers improve and produce finer-grained continuous scores, the LP's edge over thresholding grows: it exploits the full continuous surface, while a threshold collapses increasingly precise scores into a binary split.
> > >
> > > These are not post-hoc labels. Each one follows from the game structure, and removing any one would lose a measurable part of the LP's advantage.
> > >
> > > The revision will present the game-theoretic formulation as central to the contribution. We note that the original review characterized the method as "constrained reranking rather than game-theoretic" and suggested the framing overstated novelty. We revised accordingly. The follow-up now suggests this reframing weakens novelty. We have tried to engage constructively with both directions and have provided new experimental evidence. We would welcome specific suggestions on how the reviewer believes the framing should be adjusted, so we can act on them in the revision.

---

### Decision · Program_Chairs · 2026-04-30

**Decision:**

Accept (regular)

**Comment:**

This paper proposes Safety Game, a black-box inference-time framework for LLM safety alignment.
Given a set of candidate responses, the method uses self-probing to estimate helpfulness and safety risk scores, then solves a linear program derived from a two-player zero-sum game formulation to select the optimal response within a configurable risk budget. The framework does not require model weights, retraining, or internal access.

All four reviewers find the problem important and practically relevant. Reviewer qaiR and aChB highlight the clean separation of measurement from decision-making, noting that scorer improvements plug directly into the LP without modification. The black-box constraint has clear deployment advantages for organizations relying on API-only access.

The initial rebuttal added a Llama-Guard comparison on SORRY-Bench. On identical Llama-Guard scores, the LP improved all three metrics over filter-then-rank and offered a better quality-coverage tradeoff than safety-only ranking. Noise injection showed graceful degradation.

Three reviewers clearly raised their scores during the discussion period. Reviewer aChB moved from 2 to 4. Reviewer S9AW raised to 5 in the discussion after the threshold experiment, noting it "successfully demonstrates the practical difference between applying an aggregate risk constraint vs. standard individual filtering." Reviewer 3RJ5 raised their score. Reviewer qaiR maintained 5 throughout and confirmed concerns addressed. it is a successful discussion.

The discussion-period experiments resolved the central question: the LP adds concrete, structural value over simpler decision rules on identical scores. It should be accepted